# Anti-Invasion and Antiangiogenic Effects of Stellettin B through Inhibition of the Akt/Girdin Signaling Pathway and VEGF in Glioblastoma Cells

**DOI:** 10.3390/cancers11020220

**Published:** 2019-02-14

**Authors:** Shu-Yu Cheng, Nan-Fu Chen, Pi-Yu Lin, Jui-Hsin Su, Bing-Hung Chen, Hsiao-Mei Kuo, Chun-Sung Sung, Ping-Jyun Sung, Zhi-Hong Wen, Wu-Fu Chen

**Affiliations:** 1Doctoral Degree Program in Marine Biotechnology, National Sun Yat-Sen University, No. 70, Lianhai Road, Gushan District, Kaohsiung City 80424, Taiwan; joygetit@gmail.com; 2Doctoral Degree Program in Marine Biotechnology, Academia Sinica, No. 128, Section 2, Academia Rd, Nangang District, Taipei City 11529, Taiwan; 3Department of Neurosurgery, Kaohsiung Chang Gung Memorial Hospital and Chang Gung University College of Medicine, No. 123, Dapi Road, Niaosong District, Kaohsiung City 833, Taiwan; 4Division of Neurosurgery, Department of Surgery, Kaohsiung Armed Forces General Hospital, No. 2, Zhongzheng 1st Road, Lingya District, Kaohsiung City 802, Taiwan; chen06688@gmail.com; 5Department of Neurological Surgery, Tri-Service General Hospital, National Defense Medical Center, No. 325, Section 2, Chenggong Road, Neihu District, Taipei City 114, Taiwan; 6Department of Marine Biotechnology and Resources, National Sun Yat-sen University, No. 70, Lianhai Road, Gushan District, Kaohsiung City 80424, Taiwan; cynthia811122@gmail.com (P.-Y.L.); Hsiaomeikuo@gmail.com (H.-M.K.); pjsung@nmmba.gov.tw (P.-J.S.); 7Graduate Institute of Marine Biology, National Dong Hwa University, No. 2, Houwan Road, Checheng Township, Pingtung County 944, Taiwan; x2219@nmmba.gov.tw; 8Department of Biotechnology, Kaohsiung Medical University, No. 100, Shiquan 1st Road, Sanmin District, Kaohsiung City 80708, Taiwan; bhchen@kmu.edu.tw; 9Department of Medical Research, Kaohsiung Medical University Hospital, No. 100, Shiquan 1st Road, Sanmin District, Kaohsiung City 80708, Taiwan; 10The Institute of Biomedical Sciences, National Sun Yat-Sen University, No. 70, Lianhai Road, Gushan District, Kaohsiung City 80424, Taiwan; 11Center for Neuroscience, National Sun Yat-Sen University, No. 70, Lianhai Road, Gushan District, Kaohsiung City 80424, Taiwan; 12Department of Anesthesiology, Taipei Veterans General Hospital, No. 201, Section 2, Shipai Road, Beitou District, Taipei City 112, Taiwan; cssung@vghtpe.gov.tw; 13School of Medicine, National Yang-Ming University, No. 155, Section 2, Linong St, Beitou District, Taipei City 112, Taiwan; 14National Museum of Marine Biology and Aquarium, No. 2, Houwan Road, Checheng Township, Pingtung County 944, Taiwan

**Keywords:** glioblastoma multiforme (GBM), stellettin B, angiogenesis, invasion, Girdin, VEGF

## Abstract

Angiogenesis and invasion are highly related with tumor metastatic potential and recurrence prediction in the most aggressive brain cancer, glioblastoma multiforme (GBM). For the first time, this study reveals that marine-sponge-derived stellettin B reduces angiogenesis and invasion. We discovered that stellettin B reduces migration of glioblastoma cells by scratch wound healing assay and invasion via chamber transwell assay. Further, stellettin B downregulates Akt/Mammalian Target of Rapamycin (Akt/mTOR) and Signal transducer and activator of transcription 3 (Stat3) signaling pathways, which are essential for invasion and angiogenesis in glioblastoma. This study further demonstrates that stellettin B affects filamentous actin (F-actin) rearrangement by decreasing the cross-linkage of phosphor-Girdin (p-Girdin), which attenuates glioblastoma cell invasion. Moreover, stellettin B blocks the expression and secretion of a major proangiogenic factor, vascular endothelial growth factor (VEGF), in glioblastoma cells. Stellettin B also reduces angiogenic tubule formation in human umbilical vein endothelial cells (HUVECs). In vivo, we observed that stellettin B decreased blood vesicle formation in developmental zebrafish and suppressed angiogenesis in Matrigel plug transplant assay in mice. Decreased VEGF transcriptional expression was also found in stellettin B–treated zebrafish embryos. Overall, we conclude that stellettin B might be a potential antiangiogenic and anti-invasion agent for future development of therapeutic agents for cancer therapy.

## 1. Introduction

Glioblastoma multiforme (GBM), a stage IV brain tumor, is a common aggressive malignant primary cancer of the central nervous system (CNS), accounting for approximately 40% of patients with CNS cancer [1]. The 5-year survival rate is approximately 5.5% for patients with GBM undergoing standard brain cancer treatments, including combinations of radiotherapy, surgical resection, and alkylating agent treatments [2]. Usually, GBM with complex vascularization and diffusive invasion is highly related to tumor recurrence [3]. Therefore, angiogenic signaling inhibition is a crucial component of GBM treatment. Herein, we discuss the role of stellettin B, a marine-sponge-derived compound, in cell proliferation, invasion, and angiogenesis in GBM.

Stellettin B is a marine-sponge-derived triterpenoid and was first isolated by McCormick et al. [4]. To date, a few studies have indicated that stellettin B induces apoptotic cell death through inhibition of the phosphatidylinositol 3-kinase/Akt/mammalian target of rapamycin (PI3K/Akt/mTOR) signaling pathway in human cancers, as demonstrated in glioblastoma cancer SF295 cells [5], human non-small-cell lung cancer cells [6], and human chronic myeloid leukemia cells [7]. Although stellettin B is known to have anticancer activity through the inhibition of the Akt pathway, the roles of stellettin B in other crucial mechanisms, such as Akt signaling for cell mobility regulation [8] and antiangiogenesis [9], in brain cancer have not been discussed.

In brain cancer, proliferative and invasive cancer cells infiltrate normal brain tissues, increasing the malignancy of brain cancer [10]. Amplified epidermal growth factor receptor (EGFR) signaling is observed in 40%—60% of the patients with primary GBM [11]. In GBM, the EGFR-dependent signaling pathway, phosphoinositide 3-kinase (PI3K)/Akt, and signal transducer and activator of transcription-3 (Stat3) signaling are frequently overexpressed [12], which regulates cell angiogenesis, proliferation, adhesion, and migration [13]. Moreover, a high level of PI3K/Akt signaling is correlated with tumorigenesis and cancer therapy resistance, contributing to poor prognoses [14]. Cancer invasion is a complex cell migration process that involves the F-actin cytoskeleton and focal adhesion proteins [15]. One of the Akt substrate, Girders of actin filament (Girdin, also known as GIV), is an F-actin cytoskeleton binding protein, and it facilitates cytoskeleton rearrangement and cell motility at the leading edge during cell migration [16]. Girdin expression and Stat3 activation act as a positive feedback loop in the invasion of cancer cells [17]. Furthermore, Girdin has been highly correlated with poorer prognosis in patients with lung cancer [18] and human EGFR2-positive breast cancer [19]. In GBM, Girdin was reported to be a metastasis-promoting protein that directly affects invasion, migration, and adhesion [20]. Therefore, Girdin and its related molecules have become therapeutic targets for preventing cancer invasion. In this study, we observed that stellettin B inhibits Girdin phosphorylation and disrupts F-actin formation in glioblastoma cells.

Several studies have indicated that hypoxia, low tumor oxygenation, is a major concern in patients with GBM because it promotes tumor invasion into normal regions [21]. Hypoxia triggers expression of hypoxia-inducible factors (HIFs) or oxygen homeostasis regulators, which activate vascular endothelial growth factor (VEGF) for the formation of new blood vessels in several cancer types, including GBM [22]. Therefore, the inhibition of VEGF expression is a crucial strategy in GBM treatment. Clinical reports have noted that GBM has very high VEGF expression compared with low-grade brain tumors [23]. Stat3 also regulates VEGF for angiogenesis in cancer [24]. Phosphorylated Stat3 is highly expressed in between 8.6% and 83.3% of patients with GBM [25] and affects tumor angiogenesis and migration in hypoxically stressed glioblastoma [26]. Thus, VEGF-induced angiogenesis plays a critical role in brain tumor progression. Developing antiangiogenic agents that target angiogenesis inducers in VEGF-related signaling is crucial for treating GBM.

In this study, we investigated the anti-invasion and antiangiogenesis ability of stellettin B in cellular and molecular mechanisms. We demonstrated that stellettin B inhibits cell proliferation through the Akt/mTOR pathway and cell mobility through the inhibition of p-Girdin and F-actin interaction in glioblastoma cells. Moreover, stellettin B downregulates VEGF mediators, Stat3 and HIF-1, which leads to the inhibition of the expression and secretion of the major angiogenesis inducer, VEGF, in glioblastoma cell lines. Furthermore, stellettin B significantly inhibits angiogenesis in a zebrafish intersegmental vessel (ISV) angiogenesis model and Matrigel plug angiogenesis assay in mice. Therefore, we speculate that stellettin B is a potential agent of anti-invasion and antiangiogenesis in GBM.

## 2. Results

### 2.1. Cell Death Rate and Morphological Changes in Stellettin B–Treated Human Glioblastoma Cells

Human glioblastoma U87MG and GBM8401 cells were treated with various concentrations of the marine-sponge-derived compound, stellettin B (1, 5, 10, 25, 50, or 100 μM), for 24 or 48 h. The cell viability of the glioblastoma cells after stellettin B treatment was determined using 3-(4,5-dimethylthiazol-2-yl)-2,5-diphenyl tetrazolium bromide (MTT) assay (Figure 1a). Stellettin B treatment significantly reduced cell survival at doses ≥5 μM within 24 h and ≥1 μM within 48 h in the U87MG and GBM8401 cells. Microscopic observation revealed that stellettin B treatment changed the morphology of both the U87MG and GBM8401 cells (Figure 1b).

### 2.2. Stellettin B Suppresses Migration in Glioblastoma Cells

Migration is highly correlated with failed chemotherapy and irradiation in patients with GBM and invasive glioma [27]. To preliminarily investigate the effect of stellettin B on migration and invasion in glioblastoma, we used scratch wound healing and transwell migration assay, respectively. We observed that the closure rate of GBM8401 cells was significantly lower when stellettin B treatment was applied at doses of 0.5, 1.0, 2.5, and 5 μM (Figure 2a). Furthermore, transwell migration assay demonstrated that stellettin B significantly downregulated GBM8401 and U87MG cell migration (Figure 2b). Overall, these results indicated that stellettin B inhibited the migration and invasion in glioblastoma cells.

### 2.3. Stellettin B Suppresses Akt/mTOR/Girdin Signaling and Affects Cell Movement in p-Girdin/F-Actin Interaction in Glioblastoma Cell Lines

The Akt/mammalian target of rapamycin (Akt/mTOR) pathway is the most frequently mutated pathway in human cancers, including GBM, and is correlated with tumorigenesis, drug resistance, cancer progression, and transformation [28]. To assess the effect of stellettin B on the Akt/mTOR pathway, we used constitutive Akt-activated glioblastoma cell lines, U87MG and GBM8401, for the following experiments. Western blot analysis revealed that stellettin B treatment dose-dependently downregulated Akt, mTOR, and ribosomal protein S6 phosphorylation in both the U87MG and GBM8401 glioblastoma cells within 24 h (Figure 3). Akt protein was previously discovered to interact with Girdin and affect actin organization-related cell mobility [16]. In addition, we demonstrated that stellettin B inhibited invasion and migration in glioblastoma cells. The Western blot analysis showed that stellettin B significantly inhibited p-Girdin, a regulator of F-actin rearrangement, in both the U87MG and GBM8401 cells (Figure 4a). The main function of active Girdin is to interact with F-actin at cell edges to induce cell mobility. In this study, we observed that stellettin B decreased the colocalization of p-Girdin and F-actin. Moreover, stellettin B caused cell shrinkage and decreased the amount of F-actin at cell edges (Figure 4b). Collectively, the inhibition of Akt/Girdin signaling and blocking of F-actin polymerization at cell edges indirectly demonstrate the antimigration and anti-invasion effects of stellettin B in glioblastoma cells.

### 2.4. Stellettin B Suppresses VEGF Expression and Secretion through Downregulated Stat3 and HIF-1α in Glioblastoma

VEGF is a key inducer of angiogenesis and is usually upregulated in GBM [29]. VEGF is reportedly upregulated through the activation of hypoxia-inducible factor 1 (HIF-1) and signal transducer and activator of transcription 3 (Stat3) proteins [30]. Moreover, Stat3 binds with HIF-1α to promote VEGF expression in hypoxic tumors [31]. The Akt/mTOR/S6 signaling pathway is an essential regulator of HIF-1α. We observed that stellettin B inhibited Akt/mTOR signaling. Furthermore, Western blot analysis revealed that stellettin B treatment dose-dependently and significantly downregulated p-Stat3, HIF-1α, and VEGF in both U87MG and GBM8401 cells (Figure 5a). To investigate whether VEGF secreted by glioblastoma induces angiogenesis, the cultured media were collected and the VEGF concentrations of stellettin B–treated GBM8401 and U87MG cells were measured (Figure 5b). Collectively, the results demonstrated that stellettin B downregulates p-Stat3 and HIF-1α inhibits VEGF expression and secretion in glioblastoma cells.

### 2.5. Stellettin B Inhibits Capillary Structure Formation of HUVECs

Endothelial cell proliferation, migration, and elongation, which lead to new blood vessel formation, are the main processes in angiogenesis [32]. Highly vascular pericytes within tumors are an essential pathologic feature in malignant brain cancers, such as GBM [33]. Furthermore, tumor angiogenesis enhances the interaction between blood vessels and tumor cells, which causes tumor growth and metastasis [34]. To investigate the effect of stellettin B on angiogenic tubule formation, human umbilical vein endothelial cells (HUVECs) were seeded in the endothelial growth basal medium, EBM^TM^-2, with growth factors on Matrigel-coated wells. The medium was supplemented with stellettin B in concentrations of 0, 30, 60, 90, 120, or 240 nM for 16 or 24 h. Incomplete tubular structures were observed in stellettin B–treated HUVECs. Thus, stellettin B significantly inhibits tubulin formation in HUVECs, which was comparable with the control (Figure 6).

### 2.6. Stellettin B Inhibits Angiogenesis in In Vivo Zebrafish Embryogenesis and In Vivo Murine Matrigel Plug Models

The inhibition of angiogenesis with stellettin B treatment was confirmed in in vivo *Tg* (*fli1*:EGFP)^y1^ transgenic zebrafish embryos. This method was reported to be an effective model for antiangiogenetic drug discovery [35]. To confirm the antiangiogenesis ability of stellettin B, we observed the embryonic blood vessels of 72 hpf zebrafish embryos after treatment with various concentrations (0, 50, 100, and 250 nM) of stellettin B within 48 h of exposure. The zebrafish embryo death rate was 0% for 10 and 50 nM stellettin B treatment and 6% and 10% for 100 and 250 nM stellettin B treatment, respectively (Appendix A). The intersegmental vessels (ISVs) of the zebrafish were observed, and the number of partially interrupted ISVs was counted to evaluate the effect of stellettin B on angiogenesis development (Figure 7a). The quantification of complete ISVs at 72 hpf revealed complete ISVs in 100% ± 0.5%, 66% ± 9.7%, 68% ± 11.1%, and 25% ± 10.5% of ISVs treated with stellettin B concentrations of 0, 50, 100, and 250 nM, respectively (Figure 7b). Using real-time quantitative PCR analysis, we demonstrated that VEGF mRNA expression was decreased in zebrafish embryos treated with ≥50 nM stellettin B (Figure 7c). Moreover, an in vivo murine Matrigel plug model was used as an indicator of angiogenesis [36]. Matrigel plugs containing VEGF and heparin are red in color in Figure 7c, which indicates angiogenesis occurrence. In the Matrigel plug assay, stellettin B–containing plugs were pale red, which indicated a reduction of angiogenesis (Figure 7d). Furthermore, stellettin B significantly reduced the hemoglobin content of the plugs induced by VEGF and heparin. These data confirmed that stellettin B decreases VEGF expression and causes decreased VEGF expression and angiogenesis inhibition in in vivo zebrafish and murine models.

## 3. Discussion

Highly tumorigenic angiogenesis and invasion are key points related to poor prognosis and resistance to the normal first-line treatment in GBM [37]. To our knowledge, radiotherapy with the chemo-drugs, including temozolomide and loumustine, is one of the most common first-line therapeutic strategies for newly diagnosed GBM. The landmark clinical trials have indicated that the median survival of patients with newly diagnosed GBM was less than 1.5 years [38]. EGFR amplification, phosphatase and tensin homolog (PTEN) mutation and loss, the absence of isocitrate dehydrogenase (IDH) mutations, TP53 mutations, and 1p/19q codeletion are common in patients with GBM [39]. EGFR amplification and PTEN mutation are the most common mutations and are highly correlated with tumor angiogenesis in GBM [40,41]. Therefore, preventing angiogenesis is one of the most crucial strategies in GBM treatment. A study by Tang et al. indicated that stellettin B downregulates p-Akt and enhances caspase 3-dependent apoptosis in human glioblastoma SF295 cells [5]. Our results first demonstrated that stellettin B inhibits angiogenesis, proliferation, and invasion, which are essential for GBM treatment. Thus, further research is needed regarding the effects of the angiogenesis- and invasion-related molecular mechanisms of stellettin B.

Upregulation of the EGFR pathway and PTEN mutations activate phosphoinositide 3-kinase (PI3K)/Akt/mTOR signaling [42,43]. In approximately 88% of patients with GBM, spontaneous upregulation of the PI3K/Akt/mTOR pathway activates growth signals, including cell proliferation, tumor invasion, and autophagy, and prevents apoptosis [44,45]. Moreover, Akt directly interacts with Girdin, activating Girdin to induce VEGF-mediated angiogenesis [46]. Activated p-Girdin binds to F-actin, the expression of which is highly correlated with metastasis in breast cancer [47]. In normal biological functions, Girdin is generally involved in the regulation of pseudopodia extension by remodeling F-actin in the lamellipodia at the leading edges of migrating fibroblasts and endothelial cells [16]. Girdin is also an intrinsic regulator of new blood vessel formation during postnatal brain cortex development [46]. Furthermore, high expression of Girdin controls migration and positioning in the subgranular zone of immature progenitor cells at the dentate gyrus of the developing region in the brain [48]. In glioblastoma, Akt phosphorylation activates Girdin, which is highly correlated with the properties, progression, and malignancy of brain cancer stem cells [49]. Furthermore, Girdin was indicated to be a potential biomarker of patient survival prediction in colon cancer [50]. Our result shows that stellettin B inhibits the migration of glioblastoma (Figure 2). Moreover, stellettin B decreases Akt/Girdin signaling and controls cell movement through the inhibition of the binding of Girdin to F-actin (Figure 3 and Figure 4). Furthermore, we observed that F-actin accumulation at cell leading edges, which is much more dynamic in tumor cells and results in a higher mobile fraction [51], was inhibited in stellettin B–treated U87MG and GBM8401 cells (Figure 4b). Therefore, we suggest that stellettin B inhibits glioblastoma cell migration through the inhibition of Akt/mTOR signaling, causing downstream Girdin inactivation, which leads to downregulation of F-actin polymerization in glioblastoma cells.

Aberrant vascularization, tumor angiogenesis, and poor blood supply are common occurrences in a hypoxic tumor microenvironment and cause cancer progression [52]. Overexpression of VEGF plays a major role in angiogenesis and hypoxic glioblastoma [53]. Moreover, VEGF is secreted from cancer cells into endothelial cells in a hypoxic tumor microenvironment. Translation of VEGF is regulated by HIF-1α, Stat3, Ref-1/APE, and CBP/p300 [54,55]. In addition, several studies have indicated that activated-Stat3 is expressed in 66%—83% of patients with GBM [56]. Moreover, autocrine VEGF signaling not only regulates vascular homeostasis, but also stimulates endogenous VEGF expression and secretion in glioblastoma cells [57]. Tumors with high angiogenesis are a potential prognostic factor of death; thus, antiangiogenesis treatments, such as anti-VEGF drugs, are a crucial strategy for malignant cancer treatment [58]. In this study, we showed the antiangiogenetic function of stellettin B in in vivo zebrafish vessel formation and mouse Matrigel plug models (Figure 7). In the last decade, a humanized anti-VEGF monoclonal antibody (bevacizumab, Avastin^®^), an antibody-drug conjugate for angiogenesis inhibition in glioblastoma, has been used. Using an in vivo human xenograft tumor rat model, bevacizumab was shown to inhibit tumor cell invasion and cut off the blood supply [59]. A randomized clinical trial report by Wenger et al. indicated that bevacizumab prolongs progression-free survival and overall survival in GMB patients in whom radiotherapy and chemo drugs have failed [60]. Moreover, Johnson et al. reported that the introduction of bevacizumab treatment improved the median survival of patients with GBM from 9 months in 2006–2008 to 10–11 months in 2010–2012 [61]. Conversely, a report of Gramatzki et al. indicated that bevacizumab use with decreased steroid use in glioblastoma therapy did not significantly increase overall survival, with statistics compared for 2005–2009 and 2010–2014 in patients with newly diagnosed GBM [62]. To investigate the antiangiogenetic mechanism of stellettin B, we detected the protein expression of HIF-1α, p-Stat3, and the downstream angiogenesis effector, VEGF, which was inhibited by stellettin B treatment in glioblastoma cell lines (Figure 6). We conclude that stellettin B inhibits VEGF expression at the transcription level, not only attenuating the effects of VEGF, such as is achieved using anti-VEGF antibody drugs. Nevertheless, we also found that stellettin B inhibits the migration of GBM cells (Figure 2). Both Gramatzki et al. [62] and Johnson et al. [61] agreed that the main reason anti-VEGF antibody drugs do not work very well in GBM is because they involves complex gene mutations, but the mechanism is still unclear. Furthermore, Gramatzki et al. mentioned that prognostic factors, including isocitrate dehydrogenase (IDH) mutation and O6-methylguanine DNA methyltransferase (MGMT) promoter methylation, may affect bevacizumab treatment in patients with recurrent cancer. Thus, a combination of a VEGF signaling inhibitor and target drugs based on precise molecular genetic diagnoses will be a valuable strategy for treating patients with GBM. The stellettin B toxicity of IDH mutation and MGMT expression in patients with GBM could be a research direction in future studies.

## 4. Materials and Methods

### 4.1. Compound

Stellettin B was provided by Jui-Hsin Su (National Museum of Marine Biology and Aquarium, Taiwan). It was isolated from the marine sponge *Jaspis stellifera*. Stock 100 mM stellettin B was stored in DMSO at −20 °C.

### 4.2. Cell Culture

The human brain malignant glioma U87MG and GBM8401 lines were purchased from the Food Industry Research and Development Institute (Hsinchu, Taiwan). The human glioblastoma cancer cell line, U87MG, was maintained in Minimum Essential Medium (Thermo Fisher Scientific, Waltham, MA, USA). The human glioblastoma cancer cell line, GBM8401, was maintained in Roswell Park Memorial Institute medium (RPMI) 1640 medium (Thermo Fisher Scientific). Human umbilical vein endothelial cells (HUVECs) were maintained in Medium199 (Thermo Fisher Scientific) containing 20% fetal bovine serum (FBS), 1% P/S, 2.5 mL heparin, and 5 mL ECGS/500 mL. All media contained 50 U/mL penicillin, 50 mg/mL streptomycin (Thermo Fisher Scientific), and 10% heat-inactivated FBS (Thermo Fisher Scientific). Cell lines were cultured under a humidified atmosphere of 5% CO_2_ and 95% air at 37 °C. The cell lines were subcultured every 2–3 days up to the 10th passage. The cells thus obtained were used for the following experiments.

### 4.3. MTT Assay for Cell Viability Analysis

The viability of GBM8401 and U87MG cells was determined using MTT assay (Sigma-Aldrich, St. Louis, MO, USA). In a 96-well plate, 1 × 10^4^ cells were seeded, and various concentrations of stellettin B were added for 24 or 48 h. Subsequently, 20 μL of 5 mg/mL MTT was added to each well at the end of the treatment time. The cells were then incubated at 37 °C for 4 h in the dark. Supernatant was removed, and 100 μL of dimethyl sulfoxide (DMSO) was added to each well. Absorbance was read at 570 nm using an enzyme-linked immunosorbent assay (ELISA) reader (Epoch, BioTek, Winooski, VT, USA).

### 4.4. Endothelial Cell Tube Formation Assay

A total of 1.5 × 10^4^ HUVECs were added to endothelial cell basal medium-2 (EBM-2) (LONZA, Walkersville, MD, USA) with various concentrations of stellettin B in a Matrigel-coated 96-well plate. The EBM-2 medium contained 0.1% ascorbic acid, 0.4% hFGF-B, 0.1% recombinant3 insulin-like growth factor (R3-IGF-1), 0.1% GA-1000, 0.1% heparin, 0.1% human epidermal growth factor (hEGF), 0.1% VEGF, 0.04% hydrocortisone, and 2% FBS. After 16 or 24 h of treatment, capillary-like structures of HUVECs were imaged using phase-contrast microscopy Leica DMI 3000B (Leica Camera, Wetzlar, Germany). Tube length was calculated using SPOT Imaging Software.

### 4.5. Wound Healing Migration Assay

GBM8401 glioblastoma cells (5 × 10^4^) were seeded in a 12-well plate and cultured under a humidified atmosphere of 5% CO_2_ and 95% air at 37 °C. The cells attached to the bottom to form a 100% confluency monolayer. This monolayer was scraped in a straight line with a p200 pipet tip and extra cells were washed with 1× phosphate-buffered saline (PBS). Subsequently, cells were incubated with various concentrations (0, 0.5, 1, 2.5, and 5 μM) of stellettin B and imaged using phase-contrast microscopy (Leica DMI 3000B) after 6- and 24-h treatment at the site of the scratch line.

### 4.6. Transwell Chamber Invasion Assay

For invasion assay, GBM8401 and U87MG GBM cells (2 × 10^4^ cells /chamber) were seeded in 1% FBS treated with stellettin B in gelatin-coated upper 8-µm pore transwell chambers (Corning Inc., Corning, NY, USA) and in 10% FBS in the lower chambers to induce cell migration. After 24 h, the cells were washed with 1× PBS, fixed with 4% paraformaldehyde, and stained for 25 min using Giemsa stain. Cells in the upper transwell chambers were wiped using a cotton swab, and cells in the lower chambers were imaged using phase-contrast microscopy (Leica DMI 3000B).

### 4.7. Signaling Pathway Detection through Western Blotting

U87MG and GBM8401 cells were treated at the indicated concentration of stellettin B depending on the experiment. Supernatants were collected, and the cells were washed with PBS before adding radioimmunoprecipitation assay (RIPA) lysis buffer containing cOmplete ULTRA protease inhibitor cocktail tablets (Roche Diagnostics, Mannheim, Germany). The antibodies used in this study were as follows: p-Akt (1:1000, cat: 4060; Cell Signaling Technology, Danvers, MA, USA); Akt (1:1000, cat: 9272; Cell Signaling Technology); p-S6 (1:1000, cat: 2211, Cell Signaling); S6 (1:1000, cat: 2217; Cell Signaling Technology); p-mTOR (1:1000, cat: 2976; Cell Signaling Technology); mTOR (1:1000, cat: ab84400; Abcam, Cambridge, MA, USA); p-JNK (1:1000, cat: 4668; Cell Signaling Technology); JNK (1:1000, cat: 9252; Cell Signaling Technology); P62/SQSTM1 (1:2000, cat: 18420-1-AP, Proteintech, Rosemont, IL, USA); PARP (1:1000, cat: 9532; Cell Signaling Technology); and LC3 (1:1000, cat: AP1802a; Abgent, San Diego, CA, USA).

The immunoreactive bands were visualized using Immobilon Western chemiluminescent horseradish peroxidase substrate (cat: WBKLS0500; Merck, NJ, USA). Antibody to β-actin (1:2000, cat: A5441, Sigma-Aldrich, St. Louis, MO, USA) was used as a loading control. Images were obtained using the UVP BioChemi Imaging System (UVP LCC, Upland, CA, USA), and relative densitometric quantification was performed using LabWorks 4.0 software (UVP).

### 4.8. Antiangiogenesis-Related Gene Detection Using Quantitative Real-Time Polymerase Chain Reaction (PCR)

Total RNA was extracted from GBM cells and zebrafish using TRIzol Reagent (Thermo Fisher Scientific) according to the manufacturer’s instructions. The concentration, yield, and quality control indices of the total RNA were based on the absorbance at 260 and 280 nm, as determined using a spectrophotometer. Equal amounts of total RNA were reverse transcribed into single-strand cDNA using the iScriptTM cDNA Synthesis Kit. The first strand of cDNA was synthesized using 1 μg of total RNA, 4 μL of 5× iScript reaction mix (containing both oligo(dT) primers and random hexamers, reaction buffer with dNTP), iScript reverse transcriptase, and nuclease-free water in a total volume of 20 μL. The reaction was performed at 25 °C for 50 min and 42 °C for 30 min and was terminated through deactivation of the enzyme at 85 °C for 5 min. The specific primer sequences used in this study are listed in Appendix A.

### 4.9. Observation of p-Girdin Localized with F-Actin Using Confocal Microscopy

Glioblastoma cell lines were seeded on a glass slide and treated at the indicated time points with stellettin B of various concentrations depending on the experiment. The cells on the glass slide were fixed using cold methanol on ice for 15 min. Subsequently, the cells were permeabilized for 10 min with 0.1% TritonX-100 in PBS on ice and blocked for 1 h with 1% BSA and 0.1% Tween-20 in PBS at room temperature. F-actin primary antibodies (1:200) and p-Girdin (1:200, cat: 28067; IBL, Gunma, Japan) were incubated overnight at 4 °C in a refrigerator. Secondary florescence antibodies for F-actin included Rhodamine (TRITC)-conjugated AffiniPure Donkey Anti-Mouse IgG (H+L) (1:200, Code number:715-025-150; Jackson ImmunoResearch Labs, West Grove, PA, USA) and Fluorescein (FITC)-conjugated AffiniPure Donkey Anti-Rabbit IgG (H+L) (1:200, Code number:711-096-152; Jackson ImmunoResearch Labs). To avoid detection of the wavelength of the secondary florescence antibody, we incubated the nuclear staining agent, DRAQ7 (1:100, cat: ab109202, Abcam), for 60 min at room temperature. Confocal images were obtained using a Leica TCS SP5II equipped with Leica HyD (Leica Camera). The exposure times were the same for all cell samples on the same microscope slide.

### 4.10. ELISA of VEGF

U87MG and GBM8401 cells were plated on 10 cm^2^ culture dishes at 2 × 10^6^ cells/dish. Cell culture media were supplemented with various concentrations of stellettin B. VEGF secretion levels were measured according to the manufacturer’s instructions using a commercial Quantikine^®^ HUMAN VEGF ELISA kit (R&D systems, Minneapolis, MN, USA) and normalized to the protein content.

### 4.11. In Vivo Zebrafish ISV Angiogenesis Assay

The transgenic zebrafish line, *Tg* (*fli1*:EGFP)^y1^, which was obtained from the Taiwan Zebrafish Core Facility (Academia Sinica, Taipei, Taiwan), was used for the angiogenesis assay. Before the 6 h post-fertilization (hpf) stage of development, fertilized zebrafish eggs were treated with various concentrations of stellettin B in Hank’s buffer for 72 hpf. After treatment, the fish ISVs were photographed using a Leica HyD (Hybrid Detector). The exposure times were the same for all zebrafish samples on the same microscope slide. The angiogenesis inhibition rate of stellettin B in the zebrafish was calculated using the following formula: Complete rate of ISVs (%) = ISV number of stellettin B-treated zebrafish / ISV number of non-treated zebrafish × 100%. This experiment was approved by the National Sun Yat-sen University Animal Care Committee, Kaohsiung, Taiwan (approval reference #10448).

### 4.12. In Vivo Mouse Matrigel Plug Angiogenesis Assay

Various concentrations of stellettin B with 60 ng of VEGF (PeproTech, Rocky Hill, NJ, USA) and 30 U heparin in each 400 µL of Matrigel (Corning) were injected subcutaneously into the abdominal region of 6-week-old BALB/c mice. After 10 days, the mice were sacrificed and the plugs were removed and photographed. The hemoglobin in the Matrigel plugs was measured using Drabkin’s reagent assay. The animal use protocol and experiment were approved by the National Sun Yat-sen University Animal Care Committee, Kaohsiung, Taiwan (approval reference #10312).

### 4.13. Hemoglobin Measurement Using Drabkin’s Reagent

After the Matrigel plugs were homogenized, 30 µL of each sample was incubated in 100 µL of Drabkin’s solution (1 vial of Drabkin’s reagent (Sigma-Aldrich) containing 0.5 mL of 30% BrijL23 solution (Sigma-Aldrich) in 1 L of ddH_2_O). This solution was then incubated at 56 °C for 5 min. Substrate emission was measured using an ELISA plate reader at 540 nm. Hemoglobin concentration was measured using the standard curve.

### 4.14. Statistical Analysis

All quantitative data are presented as the mean ± standard error. Data were analyzed using the two-tailed Student’s t-test. *p* < 0.05 was considered statistically significant.

## 5. Conclusions

Our results demonstrated that stellettin B, a marine-sponge-derived triterpenoid, inhibits Akt/mTOR signaling and reduces the survival rate of glioblastoma cells. Stellettin B has been demonstrated to cause cell death through the inhibition of the PI3K/Akt/mTOR pathway, which generates reactive oxygen species in SF295 glioblastoma cells [5], and the induction of mitochondria-related apoptosis through the inhibition of K562 human chronic myeloid leukemia cells [7]. Moreover, stellettin B leads to cell cycle G1 phase arrest through the inhibition of cyclin D1 protein expression and induces autophagic flux in A549 lung cancer cells [6]. In particular, we identified that stellettin B inhibits the Akt/Girdin and Stat3 signaling pathways, which leads to the downregulation of invasion, migration, and angiogenesis in glioblastoma cells. Our investigation indicated that cells lose their migration ability through the inhibition of activated Girdin, thus disrupting Girdin/F-actin interaction at cell edges in glioblastoma cells. Furthermore, angiogenesis inhibition with stellettin B was observed both in vitro and in vivo and occurred through the blocking of Stat3/VEGF signaling. Therefore, we inferred that stellettin B may not only involve apoptosis, but also block the angiogenesis process in GBM cells. Furthermore, the secretion of VEGF proteins from glioblastoma cells decreases with the stellettin B treatment dose. We discovered that stellettin B decreases the amount of blood vessel formation in zebrafish and Matrigel plug angiogenesis in mice. However, the blood–brain barrier (BBB) is a major challenge when developing drugs for treating brain diseases. Molecules with a molecular weight <650 g/mol that are lipid soluble have a high efficiency of passive BBB diffusion. The BBB maintains brain homeostasis—which is controlled by interendothelial junctions, such as tight junctions, gap junctions, and adherens junctions—by allowing substances to move from the blood to the brain. The molecular weight of stellettin B is 462.63 g/mol, which meets the molecular size criterion for passive diffusion across the BBB. Therefore, our next goal is to investigate the stellettin B intake of the brain by using brain epithelial cell lines in parallel with artificial membrane permeability assay to evaluate the transmembrane transport of compounds across the BBB [63,64]. In summary, we believe that stellettin B could be a candidate compound for the inhibition of glioblastoma migration and invasion by acting through the Akt/Girdin pathway. Moreover, stellettin B downregulates angiogenesis by blocking VEGF expression and secretion.

## Figures and Tables

**Figure 1 cancers-11-00220-f001:**
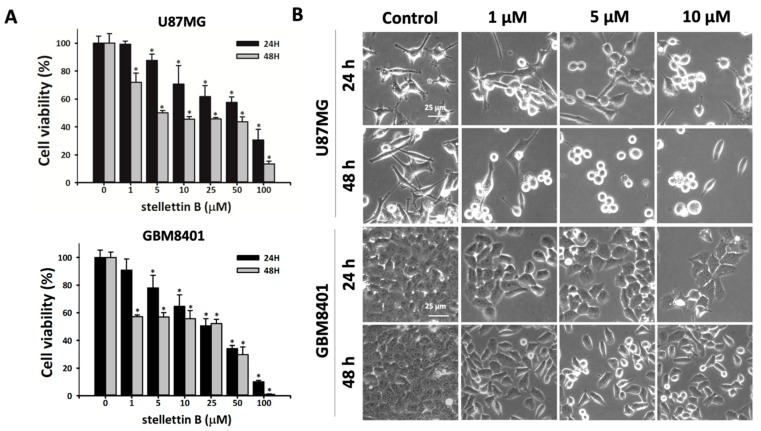
Stellettin B affects the viability of glioblastoma cell lines. (**A**) U87MG and GBM8401 cells were treated with 0, 1, 5, 10, 25, 50, or 100 µM stellettin B for 24 or 48 h. Cell viability was determined using 3-(4,5-dimethylthiazol-2-yl)-2,5-diphenyl tetrazolium bromide (MTT) assay. Data are presented as mean ± standard deviation (SD) (*n* = 3). * *p* < 0.05 relative to controls. (**B**) Morphology of U87MG and GBM8401 cells after treatment with 0, 1, 5, or 10 µM stellettin B for 24 or 48 h. Cells were observed using phase-contrast microscopy. Scale bars, 25 μm.

**Figure 2 cancers-11-00220-f002:**
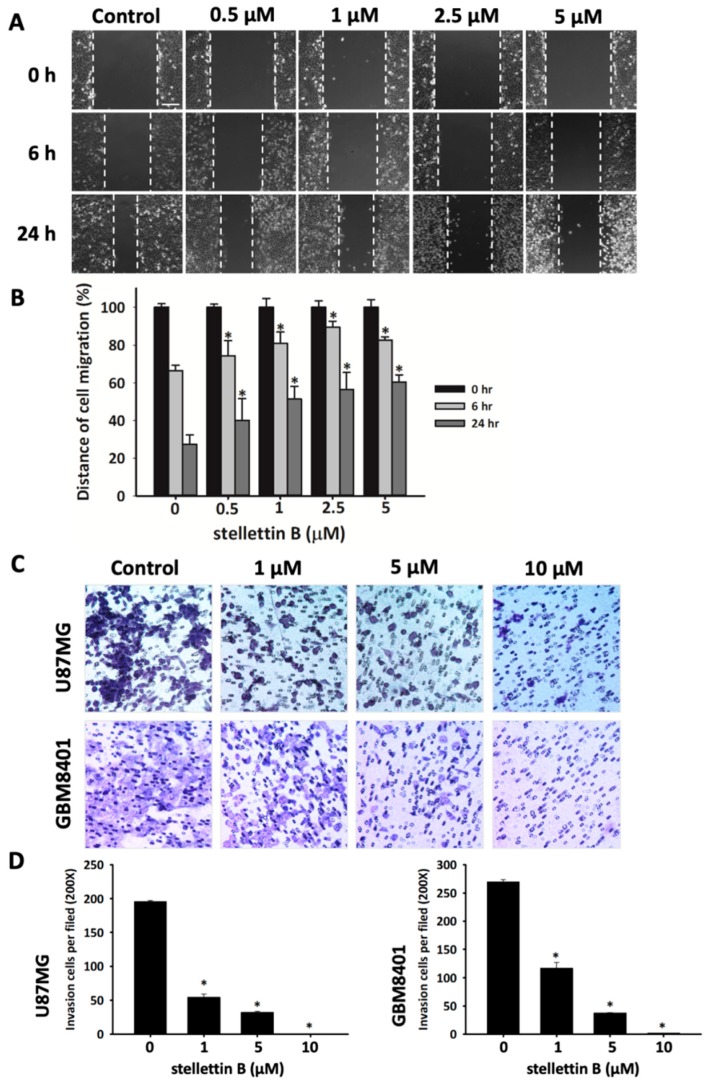
Stellettin B inhibits migration and invasion of glioblastoma (GBM) cells. (**A**) Scratch wound healing assay on GBM8401 cells treated with 0, 0.5, 1, 2.5, or 5 μM stellettin B for 6 or 24 h. Scale bar = 200 µm. (**B**) Distance of cell migration was quantified using SPOT Imaging Microscopy Imaging Software. The result is representative of three separate experiments and is presented as mean ± SD (*n* = 3). * *p* < 0.05 comparing starting time. (**C**) Cell migration was measured using a transwell chamber (8 µm pore). U87MG and GBM8401 cells were treated with 0, 1, 5, or 10 µM stellettin B for 24 h. Migrated cells were stained with Giemsa solution, magnification 200×. (**D**) The number of migrated cells on the underside of the transwell insert was counted per file. Data are presented as mean ± SD (*n* = 3). * *p* < 0.05 relative to controls.

**Figure 3 cancers-11-00220-f003:**
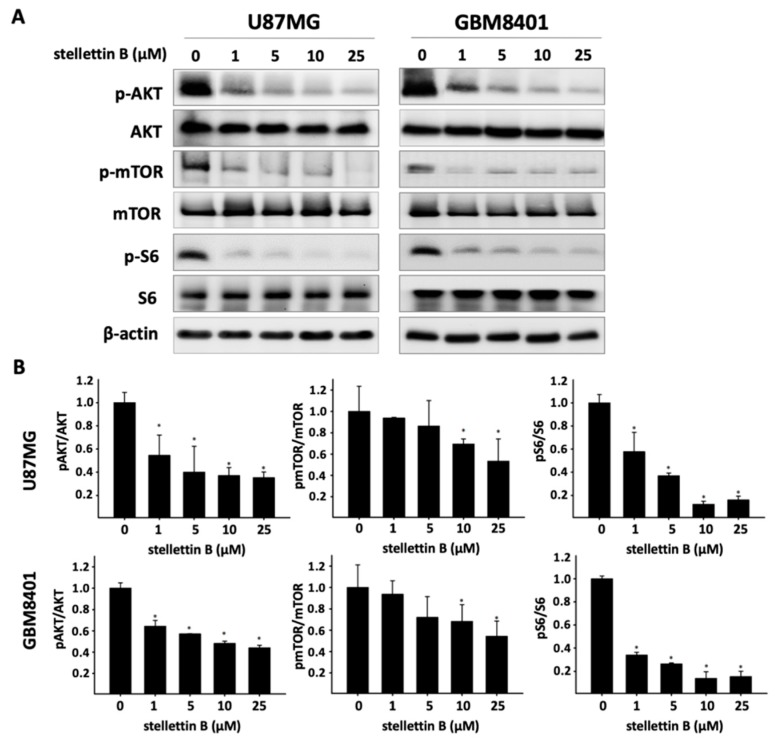
Effect of stellettin B on the Akt/mTOR pathway in glioblastoma cells. (**A**) U87MG and GBM8401 cells were treated with 0, 1, 5, 10, or 25 μM stellettin B for 24 h. Western blot analysis was performed to study the expression of p-Akt, Akt, p-mTOR, mTOR, pS6, and S6. β-Actin was used as a loading control. (**B**) p-Akt/Akt, p-mTOR/mTOR, and p-S6/S6 were quantified. Values are expressed as the mean ± SD (*n* = 3). * *p* < 0.05 relative to controls.

**Figure 4 cancers-11-00220-f004:**
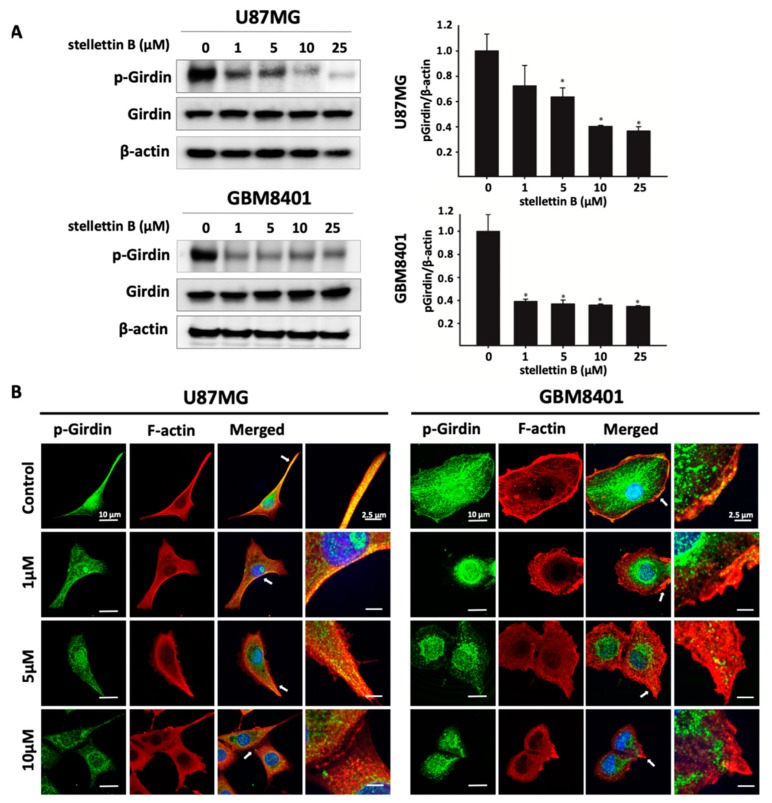
Stellettin B decreases p-Girdin and interacts with F-actin in glioblastoma cell edges. (**A**) U87MG and GBM8401 cells were treated with 0, 1, 5, 10, or 25 μM stellettin B for 24 h. Western blot analysis was performed to study the expression of p-Girdin. β-Actin was used as a loading control. p-Girdin/β-Actin were quantified. Values are expressed as the mean ± SD (*n* = 3). * *p* < 0.05 relative to controls. (**B**) Confocal images of p-Girdin (green signals), F-actin (red signals), and nuclei (blue signals) in stellettin B–treated U87MG and GBM8401 cells. Scale bars: 10 μm.

**Figure 5 cancers-11-00220-f005:**
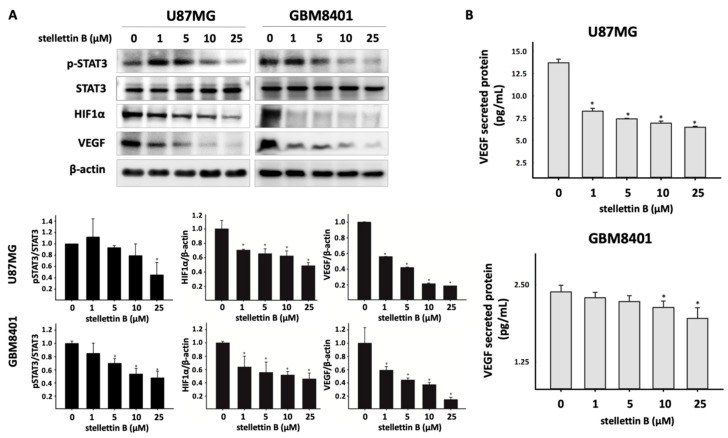
Stellettin B inhibits Stat3/HIF-1α in VEGF expression and secretion in glioblastoma cells. (**A**) U87MG and GBM8401 cells were treated with 0, 1, 5, 10, or 25 μM stellettin B for 24 h. Western blot analysis was performed to study the expression of pStat3, Stat3, HIF-1α, and VEGF. β-Actin was used as a loading control. pStat3/Stat3, HIF-1α/ β-Actin, and VEGF/ β-Actin were quantified. Values are expressed as the mean ± SD (*n* = 3). * *p* < 0.05 relative to controls. (**B**) VEGF secretion in U87MG and GBM8401 cells after 0, 1, 5, 10, or 25 μM stellettin B treatment for 24 h. Secreted VEGF was measured using the Quantikine^®^ Human VEGF immunoassay kit. Values are expressed as the mean ± SD (*n* = 4). * *p* < 0.05 relative to controls.

**Figure 6 cancers-11-00220-f006:**
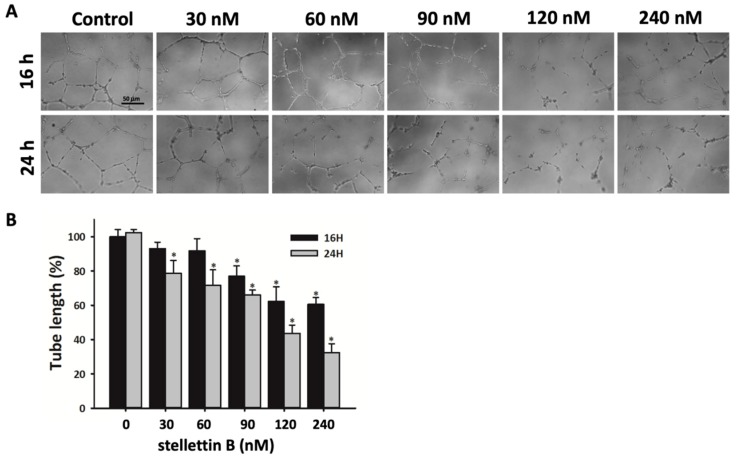
Effect of stellettin B on HUVEC tube formation. (**A**) HUVECs were plated in Matrigel-coated wells and treated with 0, 30, 60, 90, 120, or 240 nM stellettin B. Tube formation was assessed and photographed at 16 and 24 h. Scale bar = 50 µm. (**B**) The vessel length was measured for each treatment condition. Results are representative of three separate experiments and are expressed as mean ± SD (*n* = 3). * *p* < 0.05 relative to controls.

**Figure 7 cancers-11-00220-f007:**
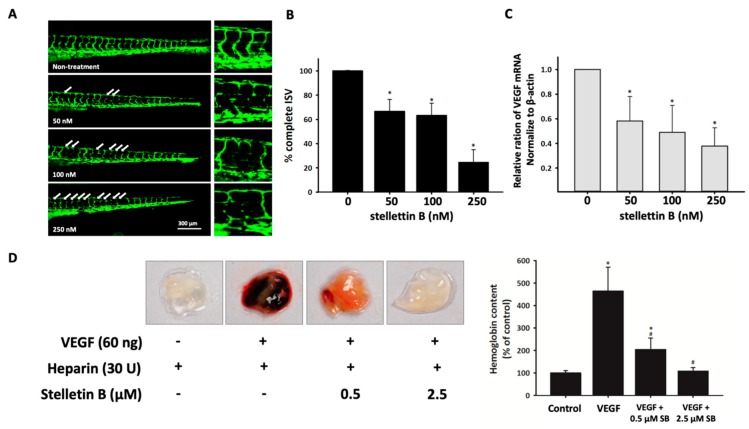
Stellettin B inhibits angiogenesis in zebrafish and mouse models. (**A**) Lateral view of *Tg* (*fli1*:EGFP)^y1^ zebrafish embryos at 72 hpf immersed in Hank’s buffer with 0, 10, 50, 100, or 250 nM stellettin B treatment. Scale bar = 300 µm. (**B**) Percentage of complete intersegmental vessels (ISVs) in each group. Values are expressed as the mean ± SD (*n* = 12). (**C**) Real-time quantitative polymerase chain reaction (PCR) analysis of VEGF expression. Values are expressed as the mean ± SD (*n* = 3). * *p* < 0.05 relative to the nontreatment group. (**D**) Effect of stellettin B on in vivo angiogenesis, determined using the Matrigel plug assay. Here, 400 µL Matrigel containing 30 U heparin with/without VEGF and/or stellettin B was subcutaneously injected into BALB/c mice. After 10 days, the Matrigel plugs were removed and photographed. The neovascularization effect on the Matrigel was evaluated through measurement of the hemoglobin content. The results are reported as the mean ± SD (*n* = 4). * *p* < 0.05 relative to the control group and # *p* < 0.05 relative to the VEGF-only group.

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
