# Peer review of "Anti-Invasion and Antiangiogenic Effects of Stellettin B through Inhibition of the Akt/Girdin Signaling Pathway and VEGF in Glioblastoma Cells"

_cancers, 2019, doi:10.3390/cancers11020220_

Round 1
Reviewer 1 Report
This article is about stellettin 3 B induced anti-invasion and antiangiogenic effects of via inhibition of the Akt/Girdin signaling 4 pathway and VEGF in U87MG and GBM8401 glioblastoma cells. Studies seems adequate somehow however, some revisions should be done.
1. In figure 1b, confluency is not correlated with figure 1a with same dose and same exposure time. For example, treatment of 10 uM of drug for 48 h to U87MG cells showed more cell number than its control group, although less than 50% survived in figure a compare to control.
2. Figure 2b, d, figure 4a right panel, figure 5a,b, figure 7 b,c are too small. Enlarge the figures.
3. Did you count more than 2500 cells per file in figure 2d with just eyes? Or did you use other method for the counting? It seems quite a work.
4. Figure 4b, are right panels (right side of merged images) enlarged images? Than another scale bar is necessary for them.
5. Figure 5a, dose unit is missing
6. Figure 6a, because of background light, it is hard to tell the difference between control and 240 nM of drug treated group for 24 h as shown in figure 6b, which is less than 40%.
7. Cytotoxicity assay in HUVECs is needed to tell the anti-tube formation effect is not due to its toxic effect. Because the sensitivity to stelletin 3B of GBM cells and HUVECs could be different.
Author Response
We would like to thank you for taking the time to review our work and for your invaluable suggestions. Based on your comments, we have made some changes to the manuscript, which are detailed below.
Comments and Suggestions for Authors This article is about stellettin 3 B induced antiinvasion and antiangiogenic effects of via inhibition of the Akt/Girdin signaling 4 pathway and VEGF in U87MG and GBM8401 glioblastoma cells. Studies seems adequate somehow however, some revisions should be done.
1. In figure 1b, confluency is not correlated with figure 1a with same dose and same exposure time. For example, treatment of 10 uM of drug for 48 h to U87MG cells showed more cell number than its control group, although less than 50% survived in figure a compare to control.
Our reply: We have found photos of U87MG cells in the same conditions from a repeated experiment we did before and replaced the figure 1b in the revised manuscript.
2. Figure 2b, d, figure 4a right panel, figure 5a,b, figure 7 b,c are too small. Enlarge the figures.
Our reply: Thank you for the suggestion. We have enlarged figure 2b, d, figure 4a right panel, figure 5a, b, and figure 7 b, c, as suggested, in the revised manuscript.
3. Did you count more than 2500 cells per file in figure 2d with just eyes? Or did you use other method for the counting? It seems quite a work.
Our reply: Yes. We did count the cells by eyes using Fiji counting tools. However, after reviewing we found the y-axis of figure 2d was incorrect. This has been corrected in the revised manuscript.
4. Figure 4b, are right panels (right side of merged images) enlarged images? Than another scale bar is necessary for them.
Our reply: Yes, they are. Thank you for noticing. We have added scale bars to the enlarged images.
5. Figure 5a, dose unit is missing
Our reply: Thank you for noticing. We have added the dose unit and enlarged the quantification results in figure 5a in the revised manuscript.
6. Figure 6a, because of background light, it is hard to tell the difference between control and 240 nM of drug treated group for 24 h as shown in figure 6b, which is less than 40%.
Our reply: We have adjusted figure 6a in the revised manuscript.
7. Cytotoxicity assay in HUVECs is needed to tell the antitube formation effect is not due to its toxic effect. Because the sensitivity to stelletin 3B of GBM cells and HUVECs could be different.
Our reply: You are right. In fact, HUVECs are more sensitive to the toxic effect of stelletin B than U87MG or GBM8401 glioblastoma cell lines are. However, the IC50 of stellettin B at 24 h is about 250 nM (HUVACs toxicity results are shown below). Therefore, we choose dosages of stelletin B that are lower than 250 nM to evaluate its effect on tube formation in HUVACs. And we found the significance dose of stellettin B for inhibiting tube formation is above 90 nM over a period of 16 h and 30 nM over a period of 24 h. Therefore, we believe that stellettin B represses the tube formation of HUVACs at the dosage that we indicated in the manuscript.
Reviewer 2 Report
Cheng et al. carried out analyses on anti-invasion and antiangiogenic effects of stellettin B through inhibition of the Akt/Girdin signaling pathway and VEGF in glioblastoma cells. The work is well written, assays correctly selected, results and discussion well presented. However, few points need to be addressed:
1) Please improve the quality for Figure’s legend as they are not readable (e.g. Figure 3,4,5,). Additionally, please list the Figure/plots with A, B and so on (Figure 3).
2) The figures should be reorganized in order to be easier to follow and draw key findings.
3) Please provide the supplier of the cell lines and MTT kit manufacturer.
4) Line 346 ‘’ A total of 1.5 × 104’’ please correct.
5) Line 354 ‘’ GBM8401 glioblastoma cells (5 × 104)’’please correct.
6) Line 358 ‘’ with various concentrations of stellettin B and imaged’’ please specify the concentration values.
7) Line 361 ‘’ GBM cells (2 × 104 cells /chamber)’’ please correct
8) Line 384/385 ‘’ TRIzol Reagent according to the 385 manufacturer’s instructions.’’Please specify the manufacturer.
9) Line 402 ‘’ 10 cm2… 2 × 106 cells/dis’’ Please correct.
10) The section for author contribution is missing.
Author Response
Comments and Suggestions
for Authors Cheng et al. carried out analyses on antiinvasion and antiangiogenic effects of stellettin B through inhibition of the Akt/Girdin signaling pathway and VEGF in glioblastoma cells. The work is well written, assays correctly selected, results and discussion well presented. However, few points need to be addressed:
We would like to thank you for carefully reviewing our work and for your invaluable suggestions. We have made some adjustments of the manuscript based on your comments and the changes are detailed below.
1) Please improve the quality for Figure’s legend as they are not readable
(e.g. Figure 3,4,5,). Additionally, please list the Figure/plots with A, B and so
on (Figure 3).
Our reply: We have enlarged Figure 2b, d, Figure 4a right panel, Figure 5a, b, and figure 7b, c. We have also remodeled Figure 3 into 3a and 3b in the revised manuscript.
2) The figures should be reorganized in order to be easier to follow and draw key findings.
Our reply: We have remodeled Figures 2, 3, 4, 5, and 7 in the revised manuscript, as suggested.
3) Please provide the supplier of the cell lines and MTT kit manufacturer.
Our reply: We have added the supplier information to Material and Method and modified the sentence as follows: “The human brain malignant glioma U87MG and GBM8401 lines were purchased from the Food Industry Research and Development Institute (Hsinchu, Taiwan).” and “The viability of GBM8401 and U87MG cells was determined using MTT assay (Sigma-Aldrich).”
4) Line 346 ‘’ A total of 1.5 × 104’’ please correct.
Our reply: Thank you for noticing this issue. We have adjusted the sentence at Line 346 to “A total of 1.5 × 104 HUVECs…”
5) Line 354 ‘’ GBM8401 glioblastoma cells (5 × 104)’’please correct.
Our reply: We have modified the information in Materials and Methods in the revised manuscript as follows: “GBM8401 glioblastoma cells (5 × 104) …”
6) Line 358 ‘’ with various concentrations of stellettin B and imaged’’ please specify the concentration values.
Our reply: We have added the information to Materials and Methods in the revised manuscript as follows: “Subsequently, cells were incubated with various concentrations (0, 0.5, 1, 2.5, and 5 μM) of stellettin B…”
7) Line 361 ‘’ GBM cells (2 × 104 cells /chamber)’’ please correct
Our reply: We have modified the information in Materials and Methods in the revised manuscript as follows: “For invasion assay, GBM8401 and U87MG GBM cells (2 × 104 cells /chamber) were…”
8) Line 384/385 ‘’ TRIzol Reagent according to the 385 manufacturer’s instructions.’’ Please specify the manufacturer.
Our reply: We have added the manufacturer information to Materials and Methods in the revised manuscript as follows: “TRIzol Reagent (Thermo Fisher Scientific, CA, USA) according to the manufacturer’s instructions.”
9) Line 402 ‘’ 10 cm2… 2 × 106 cells/dis’’ Please correct.
Our reply: We have modified the sentence in the revised manuscript as follows: “U87MG and GBM8401 cells were plated on 10 cm2 culture dishes at 2 × 106 cells/dish.”
10) The section for author contribution is missing.
Our reply: The author contribution section has been added in the revised manuscript as follows: “Author Contributions: Conceptualization, Shu-Yu Cheng and Zhi-Hong Wen; Data curation, Shu-Yu Cheng; Formal analysis, Shu-Yu Cheng, Nan-Fu Chen and Pi-Yu Lin; Funding acquisition, Nan-Fu Chen, Bing-Hung Chen, Zhi-Hong Wen and Wu-Fu Chen; Investigation, Shu-Yu Cheng and Pi-Yu Lin; Methodology, Shu-Yu Cheng, Pi-Yu Lin, Jui-Hsin Su, Hsiao-Mei Kuo, Chun-Sung Sung, Ping-Jyun Sung and Zhi-Hong Wen; Project administration, Shu-Yu Cheng, Pi-Yu Lin and Bing-Hung Chen; Resources, Jui-Hsin Su, Chun-Sung Sung, Ping-Jyun Sung, Zhi-Hong Wen and Wu-Fu Chen; Supervision, Zhi-Hong Wen and Wu-Fu Chen; Validation, Nan-Fu Chen; Writing – original draft, Shu-Yu Cheng; Writing – review & editing, Nan-Fu Chen, Zhi-Hong Wen and Wu-Fu Chen.”
Round 2
Reviewer 1 Report
Authors revised the article well and good to publish with this form.